# Outcomes in Hybrid Breast Reconstruction: A Systematic Review

**DOI:** 10.3390/medicina58091232

**Published:** 2022-09-06

**Authors:** Mario Alessandri Bonetti, Riccardo Carbonaro, Francesco Borelli, Francesco Amendola, Giuseppe Cottone, Luca Mazzocconi, Alessandro Mastroiacovo, Nicola Zingaretti, Pier Camillo Parodi, Luca Vaienti

**Affiliations:** 1Department of Reconstructive and Aesthetic Plastic Surgery, University of Milan, I.R.C.C.S. Istituto Ortopedico Galeazzi, 20161 Milan, Italy; 2Clinic of Plastic and Reconstructive Surgery, Department of Medical Area (DAME), Academic Hospital of Udine, University of Udine, 33100 Udine, Italy; 3Accademia del Lipofilling, Research and Training Center in Regenerative Surgery, 60035 Jesi, Italy; 4Clinic of Plastic and Reconstructive Surgery, c/o Ospedale “S. Maria della Misericordia”, Piazzale Santa Maria della Misericordia 15, 33100 Udine, Italy

**Keywords:** breast reconstruction, oncoplastic surgery, hybrid breast reconstruction, lipofilling, fat grafting

## Abstract

*Background and Objectives*: Lipofilling is a commonly performed procedure worldwide for breast augmentation and correction of breast contour deformities. In breast reconstruction, fat grafting has been used as a single reconstructive technique, as well as in combination with other procedures. The aim of the present study is to systematically review available studies in the literature describing the combination of implant-based breast reconstruction and fat grafting, focusing on safety, complications rate, surgical sessions needed to reach a satisfying reconstruction, and patient-reported outcomes. *Materials and Methods*: We adhered to the Preferred Reporting Items for Systematic Reviews and Meta-Analyses (PRISMA) throughout the whole review protocol. A systematic review of the literature up to April 2022 was performed using Medline, Embase, and Cochrane Library databases. Only studies dealing with implant-based breast reconstruction combined with fat grafting were included. *Results*: We screened 292 articles by title and abstract. Only 48 articles were assessed for full-text eligibility, and among those, 12 studies were eventually selected. We included a total of 753 breast reconstructions in 585 patients undergoing mastectomy or demolitive breast surgeries other than mastectomy (quadrantectomy, segmentectomy, or lumpectomy) due to breast cancer or genetic predisposition to breast cancer. Overall, the number of complications was 60 (7.9%). The mean volume of fat grafting per breast per session ranged from 59 to 313 mL. The mean number of lipofilling sessions per breast ranged from 1.3 to 3.2. *Conclusions*: Hybrid breast reconstruction shows similar short-term complications to standard implant-based reconstruction but with the potential to significantly decrease the risk of long-term complications. Moreover, patient satisfaction was achieved with a reasonably low number of lipofilling sessions (1.7 on average).

## 1. Introduction

Breast cancer is the most common malignancy in women, with a global incidence of 2,088,849 new cases and 626,679 related deaths reported in 2018 [1]. The highest incidence is predominately in Western regions including Australia, Europe, and USA [1].

The percentage of U.S. women who opted to undergo breast reconstruction after breast cancer was estimated at 43.3% based on data from NSQIP 2014 [2].

Among different types of breast reconstruction, fat grafting has been used not only as a single reconstructive technique, but also in combination with other procedures. Lipofilling might not only be useful in improving breast contour after implant-based reconstruction, but also in increasing mastectomy flap thickness prior to or associated with immediate breast reconstruction. In some cases, adjuvant lipofilling is performed after tissue expansion or at the time of expander/implant substitution, but also in autologous breast reconstruction [3].

Although initial skepticism has surrounded the oncological safety of fat grafting in breast reconstruction, studies found strong evidence demonstrating no increase in breast cancer recurrence or mortality [4,5,6,7,8].

Autologous fat transplant (AFT) has been associated with increased skin trophism and vascularization, reduced post-operative pain, and improved cosmetic results [9,10,11]. However, the debate is still ongoing over fat grafting in breast reconstruction due to some important limitations. First, the variable resorption rate makes the outcome of this procedure unpredictable. Second, the amount of fat tissue that can be grafted in a single session (especially in low volume recipient sites) is limited, which makes multiple fat grafting procedures often necessary in order to achieve satisfying outcomes [3].

Lipofilling is a commonly performed procedure worldwide for breast augmentation and correction of breast contour deformities [12]; however, heterogeneous data exist on the combination of implant-based breast reconstruction and ancillary lipofilling.

The aim of the present study is to systematically review available studies in the scientific literature describing the combination of implant-based breast reconstruction and fat grafting, focusing on safety, complications rate, number of sessions needed to reach a satisfying result (both from an aesthetic and volumetric point of view), and patient-reported outcomes.

## 2. Materials and Methods

The Preferred Reporting Items for Systematic Reviews and Meta-Analyses (PRISMA) guideline was followed throughout the design, implementation, analysis, and reporting of this systematic review [13]. Two review authors (MAB and RC) independently extracted the data from eligible studies using a standardized data extraction form, which recorded the following: study reference details (study title, authors, and year of publication); description of the patient characteristics (age, type of surgery, and length of follow-up); selection criteria; number of patients and/or breasts included in the study; outcomes of interest of the study (complications, lipofilling volumes, lipofilling sessions, and lipofilling timing with respect to implant reconstruction). Data were then transferred to a spreadsheet (Microsoft Office Excel, MS Excel, Microsoft, Redmond, Washington, USA). In case of differences in article selection, the senior author (LV) was in charge of making the final decision. A literature search updated to April 2022 was performed using the Medline, Embase, and Cochrane Library databases. The following search terms were used: (combined OR hybrid OR fat grafting OR lipofilling OR fat transfer) AND (breast reconstruction OR breast oncologic surgery). Studies found manually or through the reference lists of included studies were also eligible for inclusion. A flow chart summarizing the search strategy is depicted in Figure 1. We included studies describing implant-based breast reconstruction combined with fat grafting after complete or partial surgical breast resection due to breast cancer (i.e., therapeutic) or prophylactic. Exclusion criteria were non-English literature, flap-based reconstruction, no clearly stated follow-up, no clearly stated complications, and studies in which data on implant-based reconstructions associated with fat grafting were not extractable. Our review included case-control studies, and case series and retrospective studies; and for this reason, we assigned a level of evidence of 3. We classified complications into major, requiring implant removal (TE or implant infections, extensive flap or skin necrosis, or implant rupture); and minor, not requiring reinterventions (hematomas, seromas, fat necrosis, minor dehiscence, or implant malposition).

## 3. Results

The research produced 8924 articles, updated to April 2022. The flow chart of the study selection is outlined in Figure 1.

Initially, 292 articles were screened by title and abstract. However, only 48 articles were assessed for full-text eligibility, and among those, 12 studies were eventually included [14,15,16,17,18,19,20,21,22,23,24]. Of the 36 articles excluded after full-text reading, the majority were rejected due to a lack of clear extractable data on hybrid breast reconstruction and mentioned complications.

A total of 753 breast reconstructions in 585 patients undergoing mastectomy or demolitive breast surgeries other than mastectomy (quadrantectomy, segmentectomy, lumpectomy) due to breast cancer or genetic predisposition to breast cancer (e.g., BRCA mutation) were included. Table 1 summarizes the demographic characteristics, type of demolitive surgery, and reconstructive modalities.

The mean volume of fat grafting per breast per session ranged from 59 to 313 mL. The mean number of lipofilling sessions per breast ranged from 1.3 to 3.2 (total range 1–5). The mean follow-up was 17.8 months (range 6–48.5 months). Correlations between volume of fat grafted and complications, as well as correlations of type of complications with injected volume and with injection timing (with respect to radiotherapy) could not be statistically analyzed due to small sample sizes, nor could nonstandardized treatments with variable patient characteristics, timing, or reconstruction technique. Among the different techniques of manipulation of the harvested adipose tissue, Coleman’s technique was the most common, being used in 8 out of 12 studies; and, in all studies, adipose tissue injection was performed with 1 to 2.5 mL Luer-lock syringes and a blunt tip one-hole concave Cannula (Coleman Infiltration Cannula).

Overall complications were 60 (7.9%), of which 19 (2.5%) were classified as major complications (requiring implant removal) and 41 (5.4%) were classified as minor complications (not requiring reintervention). Among total complications, the most common was cystic fat necrosis (*n* 14, 1.9%), followed by infection requiring implant removal (*n* 12, 1.6%), seroma (*n* 10, 1.3%), dehiscence (*n* 7, 0.9%), mastectomy skin necrosis (*n* 6 (2 requiring implant removal), 0.8%), implant malposition (*n* 2, 0.3%), minor pneumothorax (*n* 1, 0.1%), and implant rupture (*n* 1, 0.1%). The capsular contracture rate was not clearly stated by some studies; for this reason, it was separately analyzed from other complications. It was mentioned in 6 out of 12 studies included for a total of 17 cases out of 337 reconstructed breasts (5%).

Variability was found in terms of postmastectomy timing of fat grafting. When radiotherapy was necessary, the time of lipofilling ranged from 3 to 19.6 months after the end of radiotherapy. However, no significant correlation was found between the timing of fat grafting and complications. The results are summarized in Table 2 and Table 3.

## 4. Discussion

Microsurgical tissue transfer is generally considered the gold standard for breast reconstruction in the case of a previously irradiated breast or anticipated necessity for breast irradiation. Furthermore, recent studies have found that autologous reconstruction yields a higher satisfaction with overall outcome and breast [25,26,27]. However, implant-based breast reconstruction is the most common reconstructive procedure, because it involves fewer scars, no donor site morbidity, and less operating time; and it does not require microsurgical skills [21,28]. It is well-known that prosthetic reconstruction is usually avoided in previously radiated breasts; in fact, alloplastic reconstruction in radiated patients carries an increased rate of both poor aesthetic outcomes and short- and long-term complications including pain, capsular contracture, and thinning of the skin, possibly resulting in visible prosthesis, implant deflation, and rupture [17,29,30,31,32,33]. Specifically, capsule contracture represents the most frequent complication experienced in radiated patients reconstructed with implants [17,31,34,35,36]. Ribuffo et al. showed that lipofilling has protective properties, allowing immediate prosthetic breast reconstruction in the setting of postoperative radiotherapy with a significantly lower complication rate [37].

Fat grafting represents another reconstructive option after breast cancer surgery, but it is usually indicated for small-volume breasts or in partial resections such as quadrantectomies and lumpectomies [3]. However, fat grafting has also been advocated as adjuvant treatment in implant-based reconstruction in order to optimize the implant’s interaction with the surrounding tissue and improve acquired breast contour deformities [38]. Moreover, fat grafting has been proven effective in revitalizing the microvascular damage and interstitial fibrosis found in chronically radio-damaged tissues [17,20,39,40]; thus, it can be exploited in breast reconstruction for both its regenerative and volumetric properties, allowing for use in lower-volume implants, which remain a foreign body, reducing postmastectomy pain and improving breast contour and consistency, leading to a more natural-looking breast [10,14,16,19,20,41,42].

A hybrid breast reconstruction protocol was proven to be associated with a lower rate of capsular contracture when compared with implant reconstruction, less breast pain at long follow-up times, and lower overall rates of revision surgery compared with standard expander-implant reconstruction [43]. However, these results are tempered by the relatively short mean follow-up period.

Studies have widely demonstrated the oncologic safety of fat grafting for breast reconstruction [5,6,7,44,45].

We were not able to draw any conclusions based on our data regarding oncologic safety because, in the included studies, no mention was made of either oncologic surveillance or cancer recurrence after fat grafting. In a recent experimental study in mice models with residual breast cancer, adipose transfer did not increase tumor size, proliferation, histologic grade, or metastatic spread, and animals receiving lipofilling showed lower tumor volume and mass after fat engraftment [46].

The well-known main disadvantage of lipofilling is the unpredictable resorption rate [14,21,47]. However, Kim et al. demonstrated a mean resorption rate of 32.9% (range, 25–52%) [41]. One study found the time to reach a steady state of fat graft retention to be as long as 2.2 years [48].

The capsular contracture rate after mastectomy and radiotherapy was not clearly stated by all studies; for this reason, it was separately analyzed from other complications. Capsular contraction was mentioned in 6 out of 12 included studies for a total of 17 cases out of 337 reconstructed breasts (5%) [16,17,18,19,20,21]. Razzouk et al. reported that 11% (15/136 breasts) of the patients developed capsular contracture (follow-up 32 months), and the average satisfaction score was 4.7 on a 5-point Likert scale [16]. Salgarello et al. reported no capsular contracture above stage 1 of Baker (23 months follow-up) [17], Hammond reported three cases (11 months follow-up) (8.3%), Sarfati et al. noticed no major capsular contracture (17 months follow-up) [19], Serra-Renom et al. reported that none of the patients in their cohort presented capsules around the prosthesis, and the Baker’s stage was never higher than one (6 months’ follow-up) [20], and Stillaert reported that no patient showed signs of capsular contracture at 24 months’ follow-up [21]. The overall rate of capsular contracture in studies reporting data on it was 5%, which is significantly lower than the contracture rate after standard implant-based breast reconstruction. Capsular contracture after breast augmentation and reconstruction affects up to 30% of patients [49]. Hammond et al. reported an overall capsular contracture incidence of 9.8%; the rate after postmastectomy radiation therapy (PMRT) was 18.7%, and 7.5% for patients without PMRT. The recent evidence suggests that periprosthetic fat grafting may decrease the capsular contracture rate [49,50,51,52,53].

Although radiotherapy has been proposed as the most important factor associated with the number of lipofilling sessions needed to complete a breast reconstruction and with the rate of complications [54], on the basis of our review, we were not able to stratify the effects of radiotherapy among complications rate or number of sessions needed to achieve satisfying results.

Gronovich et al. [11] and Hammond et al. [18] reported on direct-to-implant reconstruction with acellular dermal matrix (ADM) and tissue expander/implant reconstruction with ADM, respectively. Nothing remarkable was noted in regard to complications, capsular contracture, or aesthetic outcome. No cases of breast implant-associated anaplastic large cell lymphoma (BIA-ALCL) were reported in the included studies. This can be explained by the relatively recent introduction of the hybrid reconstructive technique and the low incidence of BIA-ALCL.

In the reviewed studies, high patient satisfaction was achieved with a reasonably low number of lipofilling sessions, averaging 1.7 sessions (range from 1.3 to 3.2 in different studies) with studies using the Likert scale (4–4.8 mean as rated by the patient, a surgeon, and a nurse on a 1–5 scale). In a study by Cigna E et al. using the VAS scale (range 1–10), the average patient satisfaction went up from 5.2 to 7.9 and surgeon satisfaction went up from 4.9 to 7.7. Studies using BREAST-Q reconstruction questionnaire reported “high” to “very-high” levels of satisfaction with final outcome (Table 1). These encouraging results are further supported by the findings of Cogliandro et al., showing that BREAST-Q was significantly better in patients who underwent hybrid breast reconstruction compared with patients who had standard implant-based reconstruction, ameliorating the cosmetic outcome as well as decreasing postoperative pain [22].

A further role of fat grafting may be related to the ability to expand indications for prepectoral breast reconstruction. The prepectoral approach in breast reconstruction is often avoided due to the increased risk for mastectomy flap necrosis and contour deformities [21]. However, fat grafting can potentially improve not only the flap thickness and vascularity but also breast contour. Consequently, it can reduce the risk of complications associated with the prepectoral placement of the implant.

Nonetheless, the combined reconstructive approach has some relevant limitations to be considered. First, several surgical procedures are often necessary to achieve final results, increasing the costs of the reconstruction. Second, it delays the psychological acceptance of the reconstructed breast [19].

Despite the promising results available in the literature and widespread use of hybrid reconstruction techniques in daily practice, our study is subjected to some limitations. First, we have to highlight the lack of comparable interstudy data (e.g., lack of report of standard deviation of the mean lipofilling sessions and injected volume) and heterogeneity in patient-reported outcomes (e.g., only two of the included studies [17,22] reported BREAST-Q scores, not all studies reported data on capsular contracture, and none reported data on oncologic safety or cancer recurrence after fat grafting). Thus, we were not able to perform any statistical analysis. Second, in the studies included, no clearly stated correlation between the abovementioned results and breast radiation therapy existed. Furthermore, a lack of information regarding pre- and postoperative mastectomy flap quality, type of implant, and implant position were noted. We therefore encountered a significant lack of high-quality prospective trials demonstrating the superiority of the hybrid breast reconstruction compared with standard implant-based reconstruction, and which is the optimal timing for autologous fat transfer (AFT).

To the best of our knowledge, this is the first systematic review of the literature on the use of implant-based reconstruction associated with fat grafting. Despite this limitation, our study demonstrates efficacy of the use of fat grafting in addition to prosthetic breast reconstruction. However, we confirmed the previously reported lack of evidence in patient-reported quality of life, and we point to the need for a high-quality randomized prospective trial comparing hybrid and standard reconstruction [55,56].

## 5. Conclusions

Hybrid breast reconstruction appears to be a minimally invasive, safe, easy-to-perform, and effective choice in breast reconstruction. Fat grafting may represent a valuable tool in plastic surgeons’ hands in reducing the risk of long-term complications, such as capsular contracture. It showed the capability to restore the thickness and trophicity of the mastectomy flap, allowing implant-based reconstruction even in the presence of poor skin quality and previously irradiated breasts.

However, the data we obtained from the included studies are tempered by the low quality of the available literature. Future larger series and randomized controlled trials are needed to confirm the results of our study and identify which patients would benefit the most from combined breast reconstruction.

## Figures and Tables

**Figure 1 medicina-58-01232-f001:**
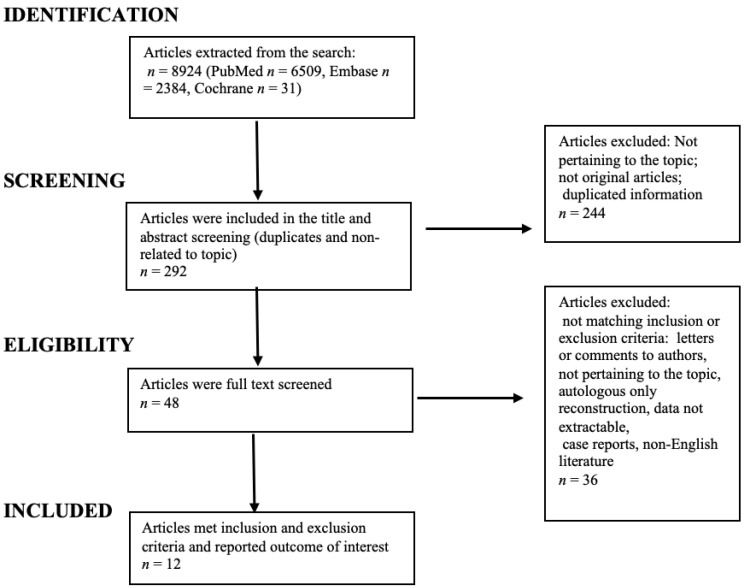
Flowchart of systematic review article selection process.

**Table 1 medicina-58-01232-t001:** Demographic characteristics.

Authors	Type of Study	N pts	N Breasts	Age	Follow-Up (Months)	Demolitive Surgery	Reconstruction Modalities	RT
Stillaert FBJL et al. (2020) [21]	Case series (single surgeon’s experience)	33	56	42 (21–77)	24.1	36 prophylactic mastectomies brca, 7 mastectomies for breast ca, 13 secondary reconstructions	step1 te; step2 (8 weeks) fat grafting; step3 prepectoral implant	pmrt in 1/33 patients
Razzouk K et al. (2019) [15]	Case Series	32	32	50.6 (9.7)	22	32 radical mastectomies followed by pmrt	step1: lipofilling (3 months post rt), step2 prepectoral implant reconstruction	32/32 pmrt
Sommeling CE et al. (2017) [14]	Case series	15	23	46 (24–64)	33	8pts (16) bilateral prophylactic mastectomies brca, 6 secondary reconstructions, 1 primary mastectomy	step 1: te, step 2 lipofilling (8 weeks post te or 6 months after pmrt in 6 pts), step3 prepectoral implant	6/23 pmrt
Razzouk K et al. (2020) [16]	multicenter retrospective study (radical mastectomy)	136	136	52 (33–72)	32.4	all modified radical mastectomies and external chest wall irradiation	step1: lipofilling (>3 months post rt), step2: prepectoral implant	136/136 pmrt
Sarfati I et al. (2011) [19]	Case series (mastectomy + radiotherapy)	28	28	45 (29–61)	17	all mastectomies for invasive breast ca and external chest wall rt	step1: lipofilling (>6, mean 9 months post rt), step2: (after mean 6.7 months.) dual plane implant reconstruction implant	28/28 pmrt
Salgarello M et al. (2012) [17]	Case series (radiotherapy + implant based reconstruction)	16	16	41 (29–58)	23.6	5 quadrantectomy/lumpectomy 2 mastectomy and pmrt, 9 bresat conserving surgeryall underwent external chest irradiation	step 1: fat grafting (all > 6 months post rt) step 2: 5 bilateral breast aug dual plane11 implant-reconstruction dual plane	16/16 pmrt
Serra-Renom JM et al. (2010) [20]	Case serie mastectomy + radiotherapy	65	65	65 (34–62)	6	all mastectomies for invasive breast ca and external chest wall rt	step1: subpectoral te and fat grafting (>1 year after radiation)step2: implant reconstruction and fat grafting	65/65 pmrt
Cigna E et al. (2012) [24]	Case series (nipple sparing)	20	20	65 (29–72)	12	nipple sparing, skin sparing and skin reducing mastectomies.	step1 implant reconstructionstep 2 fat grafting	no pmrt
Cogliandro A et al. (2017) [22]	case-control (implant based vs hybrid reconstruction)	46	46	41 (24–70)	30	mastectomy +/− external chest wall rt	step1 implant based reconstrction step 2 lipofilling (1 year after reconstruction)	34/46 pmrt
Hammond DC (2015) [18]	Case series	22	36	47 (29–66)	11	mastectomy	step 1: using te and adm step 2: fat grafting and implant	nd
Patel AA et al. (2020) [23]	Case-control (126 immediate FG, 31 delayed FG)	157 (126 IFG,31 DFG)	270 (225IFG) 45 (DFG)	46.9/49.5	48.5	ifg group (nsm 199, ssm 22 or modified rm 4)dfg goup (nsm 33, ssm 33, mrm 5)	270 breasts implant based reconstruction and immediate fat grafting45 implant reconstruction and delayed fat grafting	ifg group 20 prior rt, 28 pmrtdfggroup 9 prior rt, 4 pmrt
Gronovich Y et al. (2021) [11]	case series single surgeon (prospective)	15	25	44 (32–66)	12	mastectomywith immediate prepectoraldirect-to-implant (10 ssm 15 nsm)	all immediate implant based reconstruction with adm and immediate fat grafting	4/25 pmrt

**Table 2 medicina-58-01232-t002:** Fat grafting.

Authors	Timing of Fat Grafting	Technique	Mean Volume Per Session (mL)	Lipofilling Sessions Mean (Range)	Time between Sessions (Months)
Stillaert FBJL et al. (2020) [21]	8 weeks after onset of Tissue Expansion (or 6 months after post mastectomy RT PMRT in 1 patient)	Coleman	262	2.7 (1–5)	3
Razzouk K et al. (2019) [15]	>3 months after RT (all patients underwent PMRT)	The fat was then centrifuged 30 s at 3000 RPM	151	1.15 (1–3)	ND
Sommeling CE et al. (2017) [14]	8 weeks after onset of Tissue Expansion (or 6 months after PMRT in 6 patients)	Coleman	313	3.2 (2–5)	3
Razzouk K et al. (2020) [16]	Average time between end of radiotherapy and first lipofilling was 19.6 months	Centrifuged 30 s at 3000 revolutions	220	1.6 (1–3)	3
Sarfati I et al. (2011) [19]	Mean time was 9 months after radiotherapy.(all patients underwent PMRT)	Centrifuged speed of 3000 rpm for 3 min.	115	2 (1–3)	Mean 3.3 range (1–14)
Salgarello M et al. (2012) [17]	At least 6 months after completion of PMRT and 3 months after mastectomy if previously irradiated.	Coleman	95.7	2.4 (2–3)	>/=3
Serra-Renom JM et al. (2010) [20]	at least 1 y after mastectomy + RT. (Implant reconstruction at same time of Fat grafting)	Coleman	140	2.4 (1–4)	3
Cigna E. et al. (2012) [24]	ND	Coleman	ND	1	ND
Cogliandro A et al. (2017) [22]	1 y after implant based reconstruction (34 patients had PMRT)	Coleman	110	2.2 (1–3)	ND
Hammond DC (2015) [18]	Step 1: Using TE and ADM Step 2: Fatgrafting and implant (timing clearly stated)	The fat was strained manually of fluid	134	1.4 (1–2)	ND
Patel AA et al. (2020) [23]	126 immediate FG with Implant reconstruction, 31 delayed FG	Coleman	94	1.3 (1–2)	ND
Gronovich Y et al. (2021) [11]	At time 0 associated to prepectoral implant placement with ADM	K Vac sysrtem	59.8	1 (1)	ND

ND: not available.

**Table 3 medicina-58-01232-t003:** Complications.

Authors	N of Breast	Previous Radio Therapy	Total Complications	Minor Complications	Major Complications	Capsular Contracture
**Stillaert FBJL et al. (2020)** [21]	56	Prior RT ND, Postop adjuvant RT 1 case	4 (1 hematoma, 1 expander infection,2 implant infection)	1 (1 hematoma)	3 (1 TE infection, 2 implant infection)	0/56
**Razzouk K et al. (2019)** [15]	32	All patients underwent PMRT	5(1 implant infection, 4 cystic fat necrosis.)	4 (4cystic fat necrosis)	1 (1 Implant infection)	nd
**Sommeling CE et al. (2017)** [14]	23	PMRT in 6 patients Neoadjuvant ND	1 (1 severe infection with skin necrosis necessitated removal of the implant)	0	1 (1 implant infection with fat necrosis)	nd
**Razzouk K et al. (2020)** [16]	136	All patients underwent RT prior to Fat Grafting (mean 19.6 months)	11 (7 cystic seroma, 1 minor pnx, 1 infection (implant explantation), 2 skin necrosis(implant explantation)	7 (cystic seromas)	3 (1implant infection, 2 skin necrosis)	15/136
**Sarfati** I **et al. (2011)** [19]	28	All patients underwent RT prior to Fat Grafting (mean 9 months)	4 (4 seromas, of whom 1 patient nedeed implant explantation)	3 (minor seromas)	1 (severe seroma needing implant removal)	0/28
**Salgarello** M **et al. (2012)** [17]	16	All patients underwent RT prior to Fat Grafting (>6 months)	No complication	0	0	0/16
**Serra-Renom** JM **et al. (2010)** [20]	65	All patients underwent RT prior to Fat Grafting (>1 year)	No complications	0	0	0/65
**Cigna E. et al. (2012)** [24]	20	No postoperative RT (previous irradiation ND)	1 (1 fat necrosis)	1 (1 fat necrosis)	0	nd
**Cogliandro** A **et al. (2017)** [22]	46	34 PMRT	2 (1 infection, 1 implant rupture)	0	2 (1 infection, 1 implant rupture)	nd
**Hammond** DC **(2015)** [18]	36	RT ND (step 1 TE + ADM step2 FG + Implant)	7 (3 capsular contracture, 1 oil cyst, 1 fat necrosis, 1 red breast, 1 dehiscence)	3 (2 fat necrosis, 1 dehiscence)	1 (1 red breast)	3/36
**Patel** AA **et al. (2020)** [23]	270	29 previous RT, 32 PMRT	20 (4 infection, 6 dehiscence, 4 seromas, 3 skin necrosis, 1 fat necrosis, 2 implant malposition)	13 (6 dehiscence, 4 seromas, 1 fat necrosis, 2 implant malposition)	7 (4 infection, 3 skin necrosis)	nd
**Gronovich Y et al. (2021)** [11]	25	No preop RT, 4 Postop RT	(5) infection 1, seroma 2, dehiscence1, flap necrosis 1	3 (2 seromas, 1 dehiscence)	2 (1 infection, 1 flap necrosis)	nd

## Data Availability

All data generated or analyzed during this study are included in this article. Further enquiries can be directed to the corresponding author.

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
