# Peer review of "Outcomes in Hybrid Breast Reconstruction: A Systematic Review"

_medicina, 2022, doi:10.3390/medicina58091232_

Round 1
Reviewer 1 Report
Introduction:
Overall well written introduction. Line 47-48 would be helpful to give some additional examples on the use of fat grafting with other reconstructive techniques. Line 55-56 hard to follow the author’s line of thinking with regards to “graftable”
Methods:
Was there a reason “implant” was not used in the search terms given this was a focus of the study? Could be important to this to make sure these are not excluded.
Could the authors also comment on what consideration was made for patient’s that had undergone radiation as this would likely have affected fat grafting outcomes.
Could the authors also describe who performed the literature review and how many authors performed article selection. What was done in instances of differences in article selection – i.e. how were differences reconciled
Results:
Table 1 would recommend adding a column for postmastectomy radiation as this is a comorbiditiy that would be good to know.
Do the authors know the interval between fat grafting sessions especially since the mean went up to 3.2 sessions.
Would define complications mentioned in the results in the methods sections (i.e. what constitutes major vs minor).
The authors state the Coleman technique was the most common for fat harvesting. How was fat primarily injected in these cases/studies?
Discussion:
Well written but would consider the addition of discussion of pre-pec vs sub-pec implant based reconstruction in the setting of fat grafting. Would also discuss the timing of fat grafting i.e at the time of implant exchange vs in delayed procedures. It might also be helpful to discuss combination of lat dorsi with implant and fat grafting.
Conclusion:
I think the conclusion would benefit from tempering. It’s a little too strong and might be extreme to suggest lipofilling significantly decreases risk of long term complications.
Author Response
Introduction
Overall well written introduction. Line 47-48 would be helpful to give some additional examples on the use of fat grafting with other reconstructive techniques. Line 55-56 hard to follow the author’s line of thinking with regards to “graftable”.
The manuscript was modified according to the reviewer’s comment. We added examples of different settings in which fat grafting is used in combination with other reconstructive procedures in breast surgery. We have also attempted to be more clear in regard to the limitations of fat grafting in this field.
Methods:
Was there a reason “implant” was not used in the search terms given this was a focus of the study? Could be important to this to make sure these are not excluded.
We decided to exclude “implant” from search terms because it yielded more than 30.000 results, while adding “breast implant” as a search term did not affect the number of studies available for full text eligibility.
Could the authors also comment on what consideration was made for patient’s that had undergone radiation as this would likely have affected fat grafting outcomes.
We agree that RT is a potentially significant bias. In fact, as we pointed out in the text, we attempted to stratify complications as well as number of lipofilling sessions and volumes required based on weather patients had undergone previous radiotherapy or not. Unfortunately, the low quality of the available literature and the lack of raw data did not allow us to stratify outcomes based on previous RT. This is one of the reasons why we emphasized in the discussion and in the conclusion paragraphs that well designed studies are necessary in order to give definitive answers about the efficacy and safety of hybrid reconstruction in patients previously subjected to radiotherapy.
Could the authors also describe who performed the literature review and how many authors performed article selection. What was done in instances of differences in article selection – i.e. how were differences reconciled.
Thank you for your comment. We added a more detailed description of authors involved in data extraction and studies selection and inclusion criteria in the materials and method paragraph.
Results:
Table 1 would recommend adding a column for postmastectomy radiation as this is a comorbiditiy that would be good to know.
We have added one column concerning RT in table 1 for the studies in which this information could be retrieved. We also specified in table 2, in case the information was available, the timing between end of the radiotherapy and fat grafting.
Do the authors know the interval between fat grafting sessions especially since the mean went up to 3.2 sessions.
Not all studies reported the interval between fat grafting sessions. However we agree this can be of interest to the author so we added to table 2 (which summarizes data regarding lipofilling) this data, when available.
Would define complications mentioned in the results in the methods sections (i.e. what constitutes major vs minor).
We added to the methods section a through description of how complications were defined and classified.
The authors state the Coleman technique was the most common for fat harvesting. How was fat primarily injected in these cases/studies?
We have now added this information in the result paragraph.
Discussion:
Well written but would consider the addition of discussion of pre-pec vs sub-pec implant based reconstruction in the setting of fat grafting. Would also discuss the timing of fat grafting i.e at the time of implant exchange vs in delayed procedures. It might also be helpful to discuss combination of lat dorsi with implant and fat grafting.
We have added in the discussion a brief paragraph regarding the prepectoral vs subpectoral implant reconstruction and how hybrid technique can possibly expand indications for prepectoral breast reconstruction by allowing to improve mastectomy flap thickness, contour and trophicity.
Conclusion:
I think the conclusion would benefit from tempering. It’s a little too strong and might be extreme to suggest lipofilling significantly decreases risk of long term complications.
We tempered the conclusion emphasizing the need for a larger number of high quality studies before drawing definitive conclusions. Nonetheless we still believe that based on available literature as well as empirical clinical evidence hybrid reconstruction appears to be safe and effective and thus should be encouraged.
Reviewer 2 Report
The paper analyzes the published papers on implant and AFG reconstruction (hybrid) . The analyzed papers are not really good. So the the outcome of the metaanalysis offers no big surprise. The conclusion that "further studies are necessary" is always right.
However, the discussion should discuss important parameters.
1. Any BIA-ALCL case or discussion?
2. Implant type? textured/smooth/anatomical/round? Are there any differences?
3. Implant position: epi/sub pectoral: any influence?
4. PMRT? What was the mean follow up on that topic? If a study says " just 8% contracture" we all know that this depends on the time frame.
5. type of mastectomy: the thicker the mastectomy flaps the nicer the reconstructive outcome. any data on mastectomy type, flap thickness/BMI prior to AFG?
6. Any influence if a mesh was used?
Author Response
The paper analyzes the published papers on implant and AFG reconstruction (hybrid) . The analyzed papers are not really good. So the outcome of the meta-analysis offers no big surprise. The conclusion that "further studies are necessary" is always right.
However, the discussion should discuss important parameters.
- Any BIA-ALCL case or discussion?
- Implant type? textured/smooth/anatomical/round? Are there any differences?
- Implant position: epi/sub pectoral: any influence?
- PMRT? What was the mean follow up on that topic? If a study says " just 8% contracture" we all know that this depends on the time frame.
- type of mastectomy: the thicker the mastectomy flaps the nicer the reconstructive outcome. any data on mastectomy type, flap thickness/BMI prior to AFG?
- Any influence if a mesh was used?
Thank you very much for your comments. We believe that in consideration of the rapid spread of hybrid reconstructive technique in the clinical practice it is of outmost importance to point out to the scientific community that more studies on this topic are needed in order to draw conclusions. Moreover we are also trying to give some suggestions to authors who may be currently working on this topic on specific areas of interest with the broader topic of hybrid reconstruction (such as outcome stratification based on previous or post-op radiotherapy)
- Unfortunately none of the studies included mentioned BIA-ALCL. This may be due to the relatively short follow up which ranged from 6 to 48 months, still too short to expect to find any ALCLs. Nonetheless it would be very interesting to observe hybrid reconstructions in the long term in order to observe if there is any effect on this. We believe in consideration of the relatively recent introduction of this techniques and low incidence of ALCL a few more years will be needed to study this phenomenon.
- Quite surprisingly only very few of the studies mentioned the type of implant used (Stillaert round smooth Motiva Implants in prepec position and Sommeling only mentioned the type of Ttissue expander CPX4 Contour Profile Tissue Expander, Mentor)
- We reported the reconstructive technique in table 1 for each study included. Unfortunately, once again, not all of studies consistently reported the specific type of reconstruction modality (Stillaert, Razzouk and Sommeling described prepectoral techniques, Sarfati and Salgarello described dual plane techniques). Therefore, it wasn’t possible to draw any conclusion regarding influence of implant position on outcomes and complications.
- Regarding capsular contracture the follow up periods are all reported in Table 1. However, we limited to a descriptive analysis, without any statistical inference or any attempt of generalization because despite looking promising the result cannot be compared due to variable follow up times, variable reconstructive modalities and variable patient characteristics. Once again this shows that more standardization and consistency is required in future studies in order to make data reliable. For sake of clarity, we now added for each study reporting the rate of capsular contracture the follow up time next to the contracture rate.
- No data on mastectomy flap quality available.
- Gronovich Y. et al [11] and Hammond DC [18] reported on direct to implant reconstruction with ADM and Tissue expander/implant reconstruction with ADM respectively. Nothing remarkable was noted in regard to complications, capsular contracture nor aesthetic outcome.
Round 2
Reviewer 2 Report
The authors nicely answered my questions but did not incorporate into the manuscript.
This should be completed.
Author Response
Dear Editor,
Thank you for considering “Outcomes in hybrid breast reconstruction:a systematic review” for publication.
Hereby we will reply on the comments of the reviewer. Changes in the manuscript are indicated in red text.
Reviewers' comments:
Reviewer 2
The authors nicely answered my questions but did not incorporate into the manuscript. This should be completed.
Thank you very much for you help in improving our study.
We have now added our answers to the manuscript and tables, when appropriate (lines 249-254, 286-287).
